# DISSECTING ADAM: THE SIGN, MAGNITUDE AND VARIANCE OF STOCHASTIC GRADIENTS

## ABSTRACT

The ADAM optimizer is exceedingly popular in the deep learning community. Often it works very well, sometimes it doesn't. Why? We interpret ADAM as a combination of two aspects: for each weight, the update direction is determined by the *sign* of the stochastic gradient, whereas the update magnitude is solely determined by an estimate of its *relative variance*. We disentangle these two aspects and analyze them in isolation, shedding light on ADAM's inner workings. Transferring the "variance adaptation" to momentum-SGD gives rise to a novel method, completing the practitioner's toolbox for problems where ADAM fails.

## 1 INTRODUCTION

Many prominent machine learning models pose empirical risk minimization problems of the form

$$\min_{\theta \in \mathbb{R}^d} \mathcal{L}(\theta) = \frac{1}{M} \sum_{k=1}^{M} \ell(\theta; x_k), \quad \text{with gradient} \quad \nabla\mathcal{L}(\theta) = \frac{1}{M} \sum_{k=1}^{M} \nabla\ell(\theta; x_k), \tag{1}$$

where $\theta \in \mathbb{R}^d$ is a vector of parameters, $\{x_1, \ldots, x_M\}$ a training set, and $\ell(\theta; x)$ is a loss quantifying the performance of parameter vector $\theta$ on example $x$. Computing the exact gradient in each step of an iterative optimization algorithm becomes inefficient for large $M$. Instead, we construct a mini-batch $\mathcal{B} \subset \{1, \ldots, M\}$ of $|\mathcal{B}| \ll M$ data points sampled uniformly and independently from the training set and compute an approximate *stochastic gradient*

$$g(\theta) = \frac{1}{|\mathcal{B}|} \sum_{k \in \mathcal{B}} \nabla\ell(\theta; x_k), \tag{2}$$

which is an unbiased estimate, $\mathbf{E}[g(\theta)] = \nabla\mathcal{L}(\theta)$. We will denote by $\sigma(\theta)_i^2 := \mathbf{var}[g(\theta)_i]$ its element-wise variances.[1]

The basic stochastic optimizer is stochastic gradient descent (SGD, Robbins & Monro, 1951) and its momentum variants (Polyak, 1964; Nesterov, 1983). A number of methods, widely-used in the deep learning community, choose per-element update magnitudes based on the history of stochastic gradient observations. Among these are ADAGRAD (Duchi et al., 2011), RMSPROP (Tieleman & Hinton, 2012), ADADELTA (Zeiler, 2012) and ADAM (Kingma & Ba, 2015).

### 1.1 A CLOSER LOOK AT ADAM

We start out from a reinterpretation of the widely-used ADAM optimizer. Some of the considerations naturally extend to ADAM's close relatives RMSPROP and ADADELTA, but we restrict our attention to ADAM to keep the presentation concise. ADAM maintains moving averages of the observed stochas-

---

[1] With $k \sim \mathcal{U}(\{1, \ldots, M\})$, the gradient $\nabla\ell_k(\theta) := \nabla\ell(\theta, x_k)$ is a random variable with mean $\mathbf{E}[\nabla\ell_k(\theta)] = \nabla\mathcal{L}(\theta)$ and variances $\mathbf{var}[\nabla\ell_k(\theta)_i] = M^{-1}\sum_{k'=1}^{M}(\nabla\ell_{k'}(\theta)_i - \nabla\mathcal{L}(\theta)_i)^2$. If the elements of $\mathcal{B}$ are drawn iid with replacement, the variance of $g(\theta)$ scales inversely with $|\mathcal{B}|$: $\sigma(\theta)_i^2 := \mathbf{var}[g(\theta)_i] = |\mathcal{B}|^{-1}\mathbf{var}[\nabla\ell_k(\theta)_i]$. This holds *approximately* for sampling *without* replacement if $|\mathcal{B}| \ll M$.

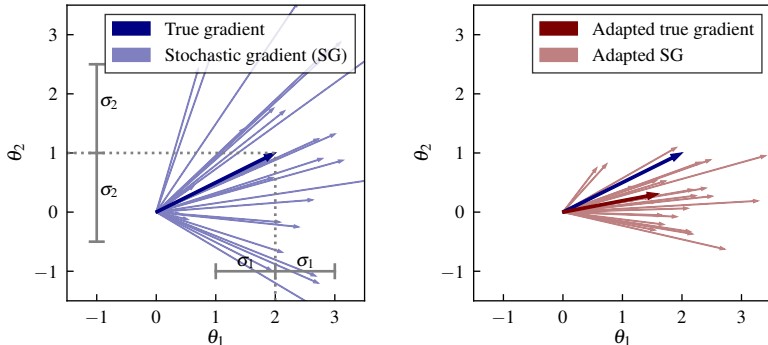

Figure 1: Conceptual sketch of variance adaptation, ignoring the sign aspect of ADAM. The left panel shows the true gradient $\nabla\mathcal{L} = (2, 1)$ and stochastic gradients scattered around it with $(\sigma_1, \sigma_2) = (1, 1.5)$. In the right panel, we employ a variance adaptation (to be derived in §3.2) that scales the $i$-th coordinate by $(1 + \eta_i^2)^{-1}$. In this example, the $\theta_2$-coordinate has much higher relative variance ($\eta_2^2 = 2.25$) than the $\theta_1$-coordinate ($\eta_1^2 = 0.25$) and is thus shortened. This reduces the variance of the update direction at the expense of biasing it away from the true gradient in expectation.

tic gradients and their element-wise square[2],

$$\tilde{m}_t = \beta_1 \tilde{m}_{t-1} + (1 - \beta_1) g_t, \qquad\qquad m_t = (1 - \beta_1^t)^{-1} \tilde{m}_t, \qquad (3)$$

$$\tilde{v}_t = \beta_2 \tilde{v}_{t-1} + (1 - \beta_2) g_t^2, \qquad\qquad v_t = (1 - \beta_2^t)^{-1} \tilde{v}_t. \qquad (4)$$

Here, $m_t$ and $v_t$ are "bias-corrected" versions of the exponential moving averages to obtain convex combinations of past observed (squared) gradients. ADAM then updates

$$\theta_{t+1} = \theta_t - \alpha \frac{m_t}{\sqrt{v_t} + \varepsilon} \qquad (5)$$

with a small constant $\varepsilon > 0$ guaranteeing numerical stability of this division. Ignoring $\varepsilon$ and assuming $|m_{t,i}| > 0$ for the moment, we can rewrite the update direction as[3]

$$\frac{m_t}{\sqrt{v_t}} = \frac{\text{sign}(m_t)|m_t|}{\sqrt{v_t}} = \frac{\text{sign}(m_t)}{\sqrt{\frac{v_t}{m_t^2}}} = \frac{\text{sign}(m_t)}{\sqrt{1 + \frac{v_t - m_t^2}{m_t^2}}}. \qquad (6)$$

Since $m_t$ and $v_t$ approximate the first and second moment of the stochastic gradient $g_t$, respectively, $v_t - m_t^2$ can be seen as an estimate of element-wise stochastic gradient variances. The division by the *non-central* second moment effectively removes the magnitude of $m_t$; it only appears in the ratio $(v_t - m_t^2)/m_t^2$. Hence, ADAM can be interpreted as a combination of the two following aspects:

- The update *direction* ($\pm$) for the $i$-th weight is given by the *sign* of $m_{t,i}$.

- The update *magnitude* for the $i$-th weight is uniquely determined by the global step size $\alpha$ and an estimate of the *relative variance*,

$$\hat{\eta}_{t,i}^2 := \frac{v_{t,i} - m_{t,i}^2}{m_{t,i}^2} \approx \frac{\sigma_{t,i}^2}{\nabla\mathcal{L}_{t,i}^2} =: \eta_{t,i}^2. \qquad (7)$$

Specifically, the update in the $i$-th coordinate is scaled by $\left(1 + \hat{\eta}_{t,i}^2\right)^{-1/2}$, shortening steps in high-relative-variance coordinates. Fig. 1 shows a sketch of this *variance adaptation*.

---

[2] Notation: Divisions, squares, etc. on vectors are to be understood *element-wise*. $\odot$ denotes element-wise multiplication. We occasionally drop $\theta$, writing $g$ instead of $g(\theta)$, etc. We use the shorthands $\nabla\mathcal{L}_t$, $g_t$, $\sigma_t^2$, etc. for sequences $\theta_t$ and double-indices, e.g. $g_{t,i} = g(\theta_t)_i$, to denote vector elements.

[3] For convenience, we define $\text{sign}(x) = 1$ for $x \geq 0$ and $\text{sign}(x) = -1, x < 0$, for our theoretical considerations, but use $\text{sign}(0) = 0$ in practice. Application to vectors is to be understood element-wise.

Table 1: The methods under consideration in this paper.

|  | **Sign + Magnitude** | **Sign** |
|---|---|---|
| **Not Variance-Adapted** | **SGD** | SSD
"Stochastic Sign Descent" |
| **Variance-Adapted** | SVAG
"Stochastic Variance-Adapted Gradient" | **ADAM** |

## 1.2 OVERVIEW

Both aspects of ADAM—taking the sign and variance adaptation—are briefly mentioned in Kingma & Ba (2015), who note that "[t]he effective stepsize [...] is also invariant to the scale of the gradients" and refer to $m_t/\sqrt{v_t}$ as a "signal-to-noise ratio". The purpose of this paper is to disentangle these two intertwined aspects in order to discuss and analyze them in isolation.

This perspective naturally suggests two alternative methods by incorporating one of the aspects but not the other (see Table 1). Taking the sign of the stochastic gradient (or momentum term) without any further modification gives rise to "Stochastic Sign Descent" (SSD). On the other hand, "Stochastic Variance-Adapted Gradient" (SVAG) applies element-wise variance adaptation factors directly on the stochastic gradient (or momentum term) instead of on its sign. We proceed as follows: In Section 2, we investigate the sign aspect. In the simplified setting of stochastic quadratic problems, we derive conditions under which the element-wise sign of a stochastic gradient can be a better update direction than the stochastic gradient itself. Section 3 discusses the variance adaptation. We present a principled derivation of "optimal" element-wise variance adaptation factors for a stochastic gradient as well as its sign. Subsequently, we incorporate momentum and briefly discuss the practical estimation of stochastic gradient variance. Section 4 presents some experimental results.

## 1.3 RELATED WORK

The idea of using the sign of the gradient as the principal source of the optimizer update has already received some attention in the literature. The RPROP algorithm (Riedmiller & Braun, 1993) ignores the magnitude of the gradient and dynamically adapts the per-element magnitude of the update based on observed sign changes. With the goal of reducing communication cost in distributed training of neural networks, Seide et al. (2014) empirically investigate the use of the sign of stochastic gradients. Regarding the variance adaptation, Schaul et al. (2013) derive element-wise step sizes for stochastic gradient descent that have (among other factors) a dependency on the stochastic gradient variance.

## 1.4 THE SIGN OF A STOCHASTIC GRADIENT

We briefly establish a fact that will be used throughout the paper. The sign of a stochastic gradient $s(\theta) = \text{sign}(g(\theta))$ estimates the sign of the true gradient. Its distribution (and thus the quality of this estimate) is fully characterized by the *success probabilities* $\rho_i := \mathbf{P}\left[s(\theta)_i = \text{sign}(\nabla\mathcal{L}(\theta)_i)\right]$. These depend on the distribution of the stochastic gradient. If we assume $g(\theta)$ to be Gaussian—which is strongly supported by a Central Limit Theorem argument on Eq. (2)—we have

$$\rho_i := \mathbf{P}\left[s(\theta)_i = \text{sign}(\nabla\mathcal{L}(\theta)_i)\right] = \frac{1}{2} + \frac{1}{2}\,\text{erf}\left(\frac{|\nabla\mathcal{L}(\theta)_i|}{\sqrt{2}\sigma(\theta)_i}\right), \tag{8}$$

see §B.2 in the supplements. Furthermore, it is $\mathbf{E}[s(\theta)_i] = (2\rho_i - 1)\,\text{sign}(\nabla\mathcal{L}(\theta)_i)$.

## 2 WHY THE SIGN?

Can it make sense to ignore the gradient magnitude? We provide some intuition under which circumstances the element-wise sign of a stochastic gradient is a better update direction than the stochastic gradient itself. This question is difficult to tackle in general, which is why we restrict the problem

class to the simple, yet insightful, case of stochastic quadratic problems, where we can investigate the effects of curvature properties and its interaction with stochastic noise.

**Model Problem** (Stochastic Quadratic Problem (QP)). *Consider the loss function $\ell(\theta, x) = 0.5\,(\theta - x)^T Q(\theta - x)$ with a symmetric positive definite matrix $Q \in \mathbb{R}^d$ and "data" coming from the distribution $x \sim \mathcal{N}(x^*, \nu^2 I)$. It is*

$$\mathcal{L}(\theta) := \mathbf{E}_x[\ell(\theta, x)] = \frac{1}{2}(\theta - x^*)^T Q(\theta - x^*) + \frac{\nu^2}{2}\operatorname{tr}(Q), \tag{9}$$

*with $\nabla \mathcal{L}(\theta) = Q(\theta - x^*)$. Stochastic gradients are given by $g(\theta) = Q(\theta - x) \sim \mathcal{N}(x^*, \nu^2 I)$.*

### 2.1 THEORETICAL COMPARISON

We want to compare update directions on stochastic QPs in terms of their expected decrease in function value from a single update step. If we update from $\theta$ to $\theta + \alpha z$, we have

$$\mathbf{E}[\mathcal{L}(\theta + \alpha z)] = \mathcal{L}(\theta) + \alpha \nabla \mathcal{L}(\theta)^T \mathbf{E}[z] + \frac{\alpha^2}{2}\mathbf{E}[z^T Q z]. \tag{10}$$

For this comparison of update *directions*, we allow for the optimal step size that minimizes Eq. (10), which is easily found to be $\alpha_* = -\nabla \mathcal{L}(\theta)^T \mathbf{E}[z]/\mathbf{E}[z^T Q z]$ and yields an expected improvement of

$$\mathcal{I}(z) := |\mathbf{E}[\mathcal{L}(\theta + \alpha_* z)] - \mathcal{L}(\theta)| = \frac{(\nabla \mathcal{L}(\theta)^T \mathbf{E}[z])^2}{2\mathbf{E}[z^T Q z]}. \tag{11}$$

We find the following expressions/bounds for the improvement of SGD and SSD:

$$\mathcal{I}(g) = \frac{1}{2}\frac{(\nabla \mathcal{L}(\theta)^T \nabla \mathcal{L}(\theta))^2}{\nabla \mathcal{L}(\theta)^T Q \nabla \mathcal{L}(\theta) + \nu^2 \sum_{i=1}^d \lambda_i^3}, \quad \mathcal{I}(s) \geq \frac{1}{2}\frac{\left(\sum_{i=1}^d (2\rho_i - 1)|\nabla \mathcal{L}(\theta)_i|\right)^2}{\sum_{i,j=1}^d |q_{ij}|} \tag{12}$$

where the $\lambda_i \in \mathbb{R}_+$ are the eigenvalues of $Q$ with orthonormal eigenvectors $v_i \in \mathbb{R}^d$. Derivations can be found in §B.1 of the supplements. Comparing these expressions, we make two observations.

Firstly, $\mathcal{I}(s)$ has a dependency on $\sum_{i,j}|q_{ij}|$. This quantity relates to the eigenvalues, as well as the orientation of the eigenbasis of $Q$. By writing $Q$ in its eigendecomposition one finds that $\sum_{i,j}|q_{ij}| \leq \sum_i \lambda_i \|v_i\|_1^2$. If the eigenvectors are perfectly axis-aligned (diagonal $Q$), their 1-norms are $\|v_i\|_1 = \|v_i\|_2 = 1$. It is intuitive that this is the best case for the intrinsically axis-aligned sign update. In general, the 1-norm is only bounded by $\|v_i\|_1 \leq \sqrt{d}\|v_i\|_2 = \sqrt{d}$, suggesting that the sign update will have difficulties with arbitrarily oriented eigenbases. We can alternatively express this matter in terms of "diagonal dominance". Assuming $Q$ has a percentage $c \in [0, 1]$ of its "mass" on the diagonal, i.e., $\sum_i |q_{ii}| \geq c \sum_{i,j}|q_{ij}|$, we can write

$$\mathcal{I}(s) \geq \frac{1}{2}\frac{\left(\sum_{i=1}^d (2\rho_i - 1)|\nabla \mathcal{L}(\theta)_i|\right)^2}{c^{-1}\sum_{i=1}^d |q_{ii}|} = \frac{1}{2}\frac{\left(\sum_{i=1}^d (2\rho_i - 1)|\nabla \mathcal{L}(\theta)_i|\right)^2}{c^{-1}\sum_{i=1}^d \lambda_i}. \tag{13}$$

Becker & LeCun (1988) empirically investigated the diagonal dominance of Hessians in optimization problems arising from neural networks and found relatively high percentages of mass on the diagonals of $c = 0.1$ up to $c = 0.6$ for the problems they investigated.

Secondly, $\mathcal{I}(g)$ contains the constant offset $\nu^2 \sum_{i=1}^d \lambda_i^3$ in the denominator, which can become hugely obstructive for ill-conditioned and noisy problems. In $\mathcal{I}(s)$, on the other hand, there is no such interaction between the magnitude of the noise and the eigenspectrum; the noise only manifests in the element-wise success probabilities $\rho_i$, its effect in the denominator is bounded. A recent paper (Chaudhari et al., 2016) investigated the eigenspectrum in deep learning problems and found it to be very ill-conditioned with the majority of eigenvalues close to zero and a few very large ones.

In summary, we can expect the sign update to be beneficial for noisy, ill-conditioned problems with "diagonally dominant" Hessians. There is some (weak) empirical evidence that these conditions might be fulfilled in deep learning problems.

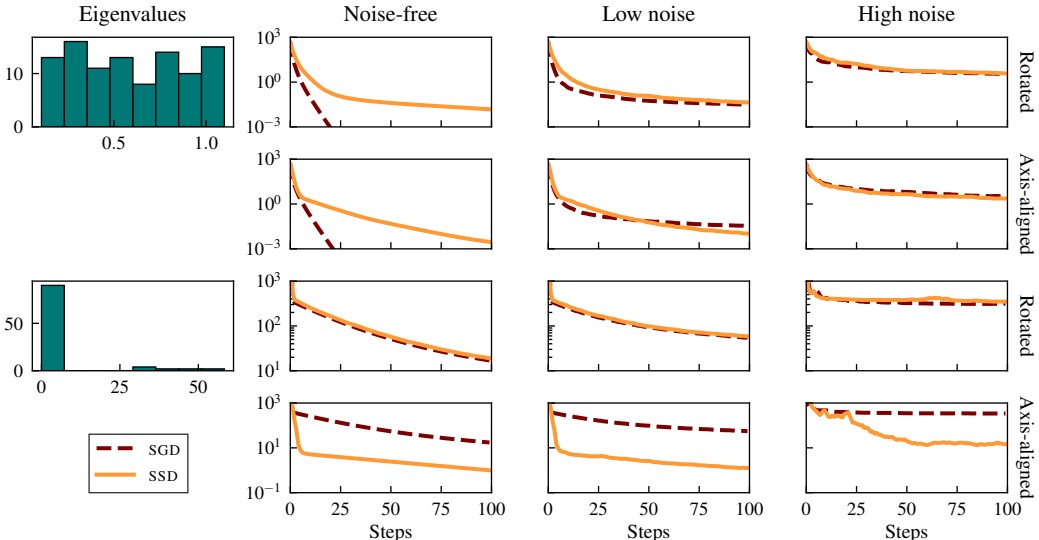

Figure 2: Performance of SGD and SSD on 100-dimensional stochastic quadratic problems. Rows correspond to different QPs: the eigenspectrum is shown and each is used with a randomly rotated and an axis-aligned eigenbasis. Columns correspond to different noise levels. Horizontal axis is number of steps; vertical axis is log function value and is shared per row for comparability.

## 2.2 EXPERIMENTAL EVALUATION

We verify the above findings on artificially generated stochastic QPs, where all relevant quantities are known analytically and controllable. We control the eigenspectrum by specifying a diagonal matrix $\Lambda$ of eigenvalues: (1) a mildly-conditioned problem with eigenvalues drawn uniformly from $[0.1, 1.1]$ and (2) an ill-conditioned problem with a structured eigenspectrum similar to the one reported for neural networks by Chaudhari et al. (2016) by uniformly drawing 90% of the eigenvalues from $[0, 1]$ and 10% from $[30, 60]$. $Q$ is then generated by (1) $Q = \Lambda$ to produce an axis-aligned problem and (2) $Q = R\Lambda R^T$ with a rotation matrix $R$ drawn uniformly at random (see Diaconis & Shahshahani, 1987). This makes four different matrices, which we consider at noise levels $\nu \in \{0, 0.1, 4.0\}$. We compare SGD and SSD, both with the optimal step size as derived from Eq. (10), which can be computed exactly in this setting.

Figure 2 shows the results, which confirm the theoretical findings. On the well-conditioned, noise-free problem, gradient descent vastly outperforms the sign-based method. Surprisingly, adding even a little noise almost evens out the difference in performance. The orientation of the eigenbasis had little effect on the performance of SSD in the well-conditioned case. On the ill-conditioned problem, the methods work roughly equally well when the eigenbasis is randomly rotated. As predicted, SSD benefits drastically from an axis-aligned eigenbasis (last row), where it clearly outperforms SGD.

## 3 VARIANCE-BASED ELEMENT-WISE STEP SIZE ADAPTATION

Besides the sign direction, the other defining property of ADAM are variance-based element-wise step sizes. Considering the variance adaptation in isolation from the sign aspect naturally suggests to employ it directly on the stochastic gradient, without taking the sign. In both cases, a motivation arises from the following consideration:

Assume we want to update in a direction $p \in \mathbb{R}^d$ (or $\text{sign}(p)$), but only have access to an unbiased estimate $\hat{p} \in \mathbb{R}^d$ with $\mathbf{E}[\hat{p}] = p$. We allow for element-wise factors $\gamma \in \mathbb{R}^d$, i.e., we update $\gamma \odot \hat{p}$ or $\gamma \odot \text{sign}(\hat{p})$. One way to make "optimal" use of these factors is to choose them such as to minimize the *expected distance* to the desired update direction. Using the squared Euclidean norm as a distance measure, we find the following result.

**Lemma 1.** *Let $\hat{p} \in \mathbb{R}^d$ be a random variable with $\mathbf{E}[\hat{p}] = p$ and $\mathbf{var}[p_i] = \sigma_i^2$. Then*

$$\min_{\gamma \in \mathbb{R}^d} \mathbf{E}[\|\gamma \odot \hat{p} - p\|_2^2] \quad \text{is solved by} \quad \gamma_i = \frac{p_i^2}{p_i^2 + \sigma_i^2} = \frac{1}{1 + \sigma_i^2/p_i^2} \tag{14}$$

*and*

$$\min_{\gamma \in \mathbb{R}^d} \mathbf{E}[\|\gamma \odot \mathrm{sign}(\hat{p}) - \mathrm{sign}(p)\|_2^2] \quad \text{is solved by} \quad \gamma_i = (2\rho_i - 1), \tag{15}$$

*where $\rho_i = \mathbf{P}[\mathrm{sign}(\hat{p}_i) = \mathrm{sign}(p_i)]$.*

In the sign case, $\gamma_i$ is proportional to the success probability with $\gamma_i = 1$ if we are certain about the sign ($\rho_i = 1$) and $\gamma_i = 0$ if we have no information about the sign at all ($\rho_i = .5$).

### 3.1 VARIANCE ADAPTATION FOR THE SIGN OF A STOCHASTIC GRADIENT

Applying Eq. (15) to $\hat{p} = g$, the optimal variance adaptation factors for the sign of a stochastic gradient are found to be $\gamma_i = 2\rho_i - 1$, where $\rho_i = \mathbf{P}[\mathrm{sign}(g_i) = \mathrm{sign}(\nabla \mathcal{L}_i)]$. Recall from Eq. (8) that, under the Gaussian assumption, the success probabilities of the sign of a stochastic gradient are $2\rho_i - 1 = \mathrm{erf}[(\sqrt{2}\eta_i)^{-1}]$. ADAM uses the variance adaptation factors $(1 + \eta_i^2)^{-1/2}$, which turns out to be a close approximation of $\mathrm{erf}[(\sqrt{2}\eta_i)^{-1}]$, as shown in Figure 5 in the supplements. Hence, ADAM can be regarded as an approximate realization of this optimal variance adaptation scheme. We experimented with both variants and found them to have identical effects. The small difference between them can be regarded as insignificant when $\eta$ itself is subject to approximation error. We thus stick to $(1 + \eta_i^2)^{-1/2}$ for accordance with ADAM and to avoid the (more costly) error function.

### 3.2 STOCHASTIC VARIANCE-ADAPTED GRADIENT (SVAG)

Applying Eq. (14) to $\hat{p} = g$, the optimal variance adaptation factors for SGD are found to be

$$\gamma_i = \frac{1}{1 + \sigma_i^2/\nabla \mathcal{L}_i^2} = \frac{1}{1 + \eta_i^2}. \tag{16}$$

This term is known from Schaul et al. (2013), where it appears together with diagonal curvature estimates in element-wise step sizes for SGD. We refer to this method (without curvature estimates) as "Stochastic Variance-Adapted Gradient" (SVAG). A momentum variant will be derived below.

Intriguingly, variance adaptation of this form guarantees convergence *without* manually decreasing the global step size. We recover the $\mathcal{O}(1/t)$ rate of SGD for smooth, strongly convex functions. We emphasize that this result considers an "idealized" version of SVAG with *exact* $\eta_i^2$. It is a motivation for this form of variance adaptation, not a statement about the performance with estimated variances.

**Theorem 1.** *Let $f$ be $\mu$-strongly convex and $L$-smooth. Assume we update $\theta_{t+1} = \theta_t - \alpha(\gamma_t \odot g_t)$, where $g_t$ is a stochastic gradient with $\mathbf{E}[g_t|\theta_t] = \nabla f(\theta_t)$, $\mathbf{var}[g_{t,i}|\theta_t] = \sigma_{t,i}^2$, variance adaptation factors $\gamma_{t,i} = (1 + \sigma_{t,i}^2/\nabla f_{t,i}^2)^{-1}$, and $\alpha = 1/L$. Assume $\mathbf{E}[\|g_t\|^2] \le G^2$. Then*

$$\mathbf{E}[f(\theta_t) - f_*] \in \mathcal{O}\left(\frac{1}{t}\right), \tag{17}$$

*where $f_*$ is the minimum value of $f$.* *(Proof in §B.4)*

### 3.3 ESTIMATING GRADIENT VARIANCE

In practice, the relative variance is of course not known and must be estimated. As noted in the introduction, ADAM obtains an estimate of the stochastic gradient variance from moving averages, $\sigma_{t,i}^2 \approx \hat{s}_{t,i} = v_{t,i} - m_{t,i}^2$. The underlying assumption is that the function does not change drastically over the "effective time horizon" of the moving average, such that the recent gradients can approximately be considered to be iid draws from the stochastic gradient distribution. An estimate of the relative variance can then be obtained by $(v_t - m_t^2)/(m_t^2)$, as in ADAM.

Unlike ADAM we do not use different moving average constants for $m_t$ and $v_t$. The constant for the moving average should define a time horizon over which the gradients can approximately be

considered to come from the same distribution. From this perspective, it is hardly justifiable to use different horizons for the gradient and its square. Furthermore, we found individual moving average constants for $m_t$ and $v_t$ to have only minor effect on the performance of our methods.

An alternative variance estimate can be computed locally "within" a single mini-batch. A more detailed discussion of both estimators can be found in §C of the supplements. We have experimented with both estimators and found them to work equally well for our purpose of variance adaptation. We thus stick to moving average-based estimates for the main paper. Appendix D provides details and experimental results for the mini-batch variant.

## 3.4 INCORPORATING MOMENTUM

When we add momentum—i.e., we want to update in the direction $r_t$ or $\text{sign}(r_t)$ with a momentum term $r_t = \mu r_{t-1} + g_t = \sum_{s=0}^{t} \mu^s g_{t-s}$—the variance adaptation factors should be determined by the relative variance of $r_t$, according to Lemma 1. It is

$$\mathbf{E}[r_t] = \sum_{s=0}^{t} \mu^s \nabla \mathcal{L}_{t-s}, \quad \mathbf{var}[r_{t,i}] = \sum_{s=0}^{t} (\mu^s)^2 \mathbf{var}[g_{t-s,i}] = \sum_{s=0}^{t} \mu^{2s} \sigma_{t-s,i}^2. \tag{18}$$

Replacing $\mathbf{E}[g_{t-s}] \approx m_{t-s}$ and $\mathbf{var}[g_{t-s}] \approx v_{t-s} - m_{t-s}^2$ we could compute these quantities. However, this would require *two* additional moving averages and can thus be discarded as impractical. Fortunately, we can motivate an approximation that does not require any additional memory requirements (see §C):

$$\frac{\mathbf{var}[r_t]}{\mathbf{E}[r_t]^2} \approx \kappa(\mu, t) \frac{v_t - m_t^2}{m_t^2} \quad \text{with} \quad \kappa(\mu, t) := \frac{(1 - \mu^{2t})(1 - \mu)^2}{(1 - \mu^2)(1 - \mu^t)^2}. \tag{19}$$

Note that the correction factor $\kappa(\mu, t)$ does not appear in ADAM, which updates in the direction $\text{sign}(m_t) = \text{sign}(r_t)$ but performs variance adaptation based on $(v_t - m_t^2)/m_t^2$. The supplements contain experiments with a variant of ADAM that includes this correction factor.

## 4 EXPERIMENTS

We compare momentum-SGD (M-SGD) and ADAM to two new methods: First, we consider M-SSD: stochastic sign descent using a momentum term. The second method is M-SVAG, i.e., SGD with momentum and variance adaptation of the form $(1 + \eta^2)^{-1}$, where the relative variance of the momentum term is estimated from moving averages according to Eq. (19). These four methods are the four possible recombinations of the sign aspect and the variance adaptation aspect of ADAM, as laid out in Table 1. Algorithms 1 and 2 provide pseudo-code for M-SSD and M-SVAG. For all experiments, we use $\mu = 0.9$ for M-SGD, M-SSD and M-SVAG and default parameters ($\beta_1 = 0.9, \beta_2 = 0.999, \varepsilon = 10^{-8}$) for ADAM. Note that M-SVAG does not use an $\varepsilon$-parameter, see Alg. 2.

---

**Algorithm 1** M-SSD (Stochastic Sign Descent with Momentum)

---

**Require:** initial value $\theta_0$, step size $\alpha$, momentum parameter $\mu \in [0, 1]$, number of steps $T$
 1: Initialize $m = 0$, $v = 0$
 2: **for** $t = 1, \ldots, T$ **do**
 3:  |  Compute stochastic gradient $g = g(\theta)$
 4:  |  Update moving average $m \leftarrow \mu m + g$
 5:  |  Update $\theta \leftarrow \theta - \alpha \, \text{sign}(m)$
 6: **end for**

---

---

**Algorithm 2** M-SVAG (Stochastic Variance-Adapted Gradient with Momentum)

---

**Require:** initial value $\theta_0$, step size $\alpha$, momentum parameter $\mu \in [0, 1]$, number of steps $T$

1: Initialize $\tilde{m} = 0$, $\tilde{v} = 0$
2: **for** $t = 1, \ldots, T$ **do**
3: $\quad$ Compute stochastic gradient $g = g(\theta)$
4: $\quad$ Update moving averages $\tilde{m} \leftarrow \mu\tilde{m} + (1 - \mu)g, \quad \tilde{v} \leftarrow \mu\tilde{v} + (1 - \mu)g^2$
5: $\quad$ Bias-correct $m = (1 - \mu^t)^{-1}\tilde{m}, \quad v = (1 - \mu^t)^{-1}\tilde{v}$
6: $\quad$ Compute relative variance estimate $\eta^2 = \kappa(\mu, t)\frac{v - m^2}{m^2}$ $\qquad\qquad\qquad$ ▷ Eq. (19)
7: $\quad$ Compute variance adaptation factors $\gamma = (1 + \eta^2)^{-1}$
8: $\quad$ Update $\theta \leftarrow \theta - \alpha(\gamma \odot m)$
9: **end for**

---

We do not use an $\varepsilon$-parameter as in ADAM. In the (rare) case that $m_i = 0$ for coordinate $i$, the division by zero in line 6 is caught and the update magnitude will be set to zero in line 8.

## 4.1 EXPERIMENTAL SET-UP

We tested all methods on three problems: a simple fully-connected neural network on the MNIST data set (LeCun et al., 1998), as well as convolutional neural networks (CNNs) on the CIFAR-10 and CIFAR-100 data sets (Krizhevsky, 2009). On CIFAR-10, we used a simple CNN with three convolutional layers, interspersed with max-pooling, and three fully-connected layers. On CIFAR-100 we used the AllCNN architecture of Springenberg et al. (2014) with a total of nine convolutional layers. A complete description of all network architectures has been moved to §A. While MNIST and CIFAR-10 are trained with a constant global step size ($\alpha$), we used a fixed decreasing schedule for CIFAR-100, dividing by 10 after 40k and 50k steps (adopted from Springenberg et al., 2014). We used a batch size of 128 on MNIST and 256 on the two CIFAR data sets.

Step sizes (initial step sizes in the case of CIFAR-100) were tuned for each method individually by first finding the maximal stable step size by trial and error, then searching downwards over two orders of magnitude (details in §A). We selected the one that yielded maximal overall test accuracy within the fixed number of training steps. Experiments with the best step size have been replicated ten times with different random seeds and all performance indicators are reported as mean plus/minus one standard deviation.

## 4.2 RESULTS

Results are shown in Figure 3. On MNIST, ADAM clearly outperforms M-SGD. Interestingly, there is only a very small difference in performance between the two sign-based methods, M-SSD and ADAM. Apparently, the advantage of ADAM over M-SGD on this problem is primarily due to the sign aspect. Going from M-SGD to M-SVAG, gives a considerable boost in performance, but M-SVAG is still outperformed by the two sign-based methods.

On CIFAR-10, the sign-based methods again have superior performance. Neither M-SSD nor M-SGD can benefit significantly from adding variance adaptation.

Finally, the situation is reversed on CIFAR-100, where M-SGD outperforms ADAM. It attains lower minimal loss values (both training and test) and converges faster. This is also reflected in the test accuracies, where M-SGD beats ADAM by almost 10 percentage points. Furthermore, ADAM is much less stable with significantly larger variance in performance. On this problem, variance adaptation has a small but significant positive effect for the sign-based methods as well as for M-SGD. When going from M-SGD to M-SVAG we gain some speed in the initial phase. The difference is later evened out by the manual learning rate decrease (which was necessary, for all methods, to train this architecture to satisfying performance).

## 5 DISCUSSION AND CONCLUSION

We have argued that ADAM combines two aspects: taking signs and variance adaptation. Our separate analysis of both aspects provides some insight into the inner workings of this method.

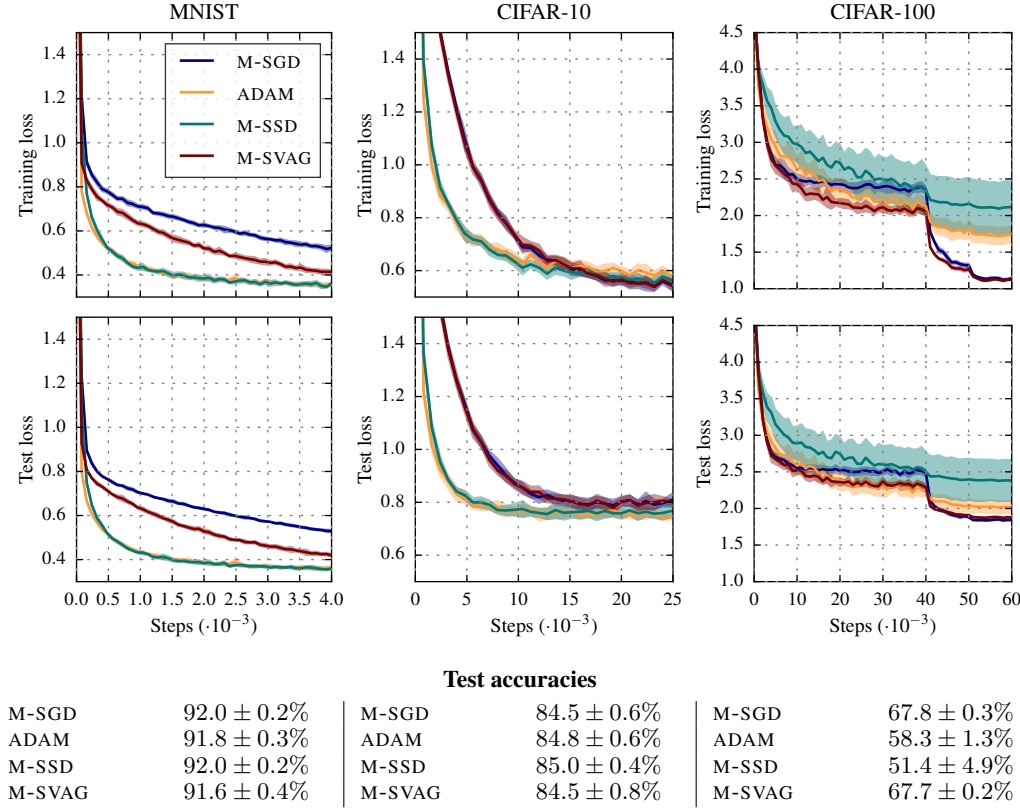

| Test accuracies | | | | | |
|---|---|---|---|---|---|
| M-SGD | $92.0 \pm 0.2\%$ | M-SGD | $84.5 \pm 0.6\%$ | M-SGD | $67.8 \pm 0.3\%$ |
| ADAM | $91.8 \pm 0.3\%$ | ADAM | $84.8 \pm 0.6\%$ | ADAM | $58.3 \pm 1.3\%$ |
| M-SSD | $92.0 \pm 0.2\%$ | M-SSD | $85.0 \pm 0.4\%$ | M-SSD | $51.4 \pm 4.9\%$ |
| M-SVAG | $91.6 \pm 0.4\%$ | M-SVAG | $84.5 \pm 0.8\%$ | M-SVAG | $67.7 \pm 0.2\%$ |

Figure 3: Experimental results on the three test problems. Plots display training and test loss over the number of steps. Curves for the different optimization methods are color-coded. The shaded area spans plus/minus one standard deviation, obtained from ten replications. The table below contains test accuracies evaluated after the last iteration.

Taking the sign can be beneficial, but does not need to be. Our theoretical analysis suggests that it depends on the interplay of stochasticity, the conditioning of the problem, and its "axis-alignment". Our experiments confirm that sign-based methods work well on some, but not all problems.

Variance adaptation can be applied to any stochastic update direction. In our experiments it was beneficial in all cases, but its effect can sometimes be minuscule. M-SVAG, a variance-adapted variant of momentum-SGD, is a useful addition to the practitioner's toolbox for problems where sign-based methods like ADAM fail. Its memory and computation cost are identical to ADAM and it has two hyper-parameters, the momentum constant $\mu$ and the global step size $\alpha$. Our TensorFlow (Abadi et al., 2015) implementation of this method will be made available upon publication.

## ACKNOWLEDGMENTS

We want to thank [names removed] for many helpful discussions.

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

SUPPLEMENTARY MATERIAL

# A  DESCRIPTION OF EXPERIMENTS

## A.1  NETWORK ARCHITECTURES

**MNIST**  We train a simple fully-connected neural network with three hidden layers of 1000, 500 and 100 units with ReLU activation. The output layer has 10 units with softmax activation. We use the cross-entropy loss function and apply $L_2$-regularization on all weights, but not the biases. We use a batch size of 128. The global learning rate $\alpha$ stays constant.

**CIFAR-10**  The CIFAR-10 data set consists of 32×32px RGB images with one of ten categorical labels. We train a convolutional neural network (CNN) with three convolutional layers (64 filters of size 5×5, 96 filters of size 3×3, and 128 filters of size 3×3) interspersed with max-pooling over 3×3 areas with stride 2. Two fully-connected layers with 512 and 256 units follow. We use ReLU activation function for all layers. The output layer has 10 units for the 10 classes of CIFAR-10 with softmax activation. We use the cross-entropy loss function and apply $L_2$-regularization on all weights, but not the biases. During training we perform some standard data augmentation operations (random cropping of sub-images, left-right mirroring, color distortion) on the input images. We use a batch size of 256. The global learning rate $\alpha$ stays constant.

**CIFAR-100**  We use the AllCNN architecture of Springenberg et al. (2014). It consists of seven convolutional layers, some of them with stride, and no pooling layers. The fully-connected layers are replaced with two layers of 1×1 convolutions with global spatial averaging in the end. ReLU activation function is used in all layers. Details can be found in the original paper. We use the cross-entropy loss function and apply $L_2$-regularization on all weights, but not the biases. We used the same data augmentation operations as for CIFAR-10 and a batch size of 256. The global learning rate $\alpha$ is decreased by a factor of 10 after 40k and 50k steps.

## A.2  LEARNING RATE TUNING

Learning rates for each optimizer have been tuned by first finding the maximal stable learning rate by trial and error and then searching downwards over two orders of magnitude with learning rates $6 \cdot 10^m$, $3 \cdot 10^m$, and $1 \cdot 10^m$ for order of magnitude $m$. We evaluated loss and accuracy on the full test set at a constant interval and selected the best-performing learning rate for each method in terms of maximally reached test accuracy. Using the best learning rate, we replicated the experiment ten times with different random seeds.

# B  MATHEMATICAL DETAILS

## B.1  DETAILS OF THE ANALYSIS ON STOCHASTIC QPS

We derive the expressions for $\mathcal{I}(s)$ and $\mathcal{I}(g)$ in Eq. (12). We drop the fixed $\theta$ from the notation for readability. For SGD, we have $\mathbf{E}[g] = \nabla\mathcal{L}$ and $\mathbf{E}[g^T Q g] = \nabla\mathcal{L}^T Q \nabla\mathcal{L} + \text{tr}(Q\mathbf{cov}[g])$, which is a general fact for quadratic forms of random variables. For the stochastic QP the gradient covariance is $\mathbf{cov}[g] = \nu^2 QQ$, thus $\text{tr}(Q\mathbf{cov}[g]) = \nu^2 \text{tr}(QQQ) = \nu^2 \sum_i \lambda_i^3$. Plugging everything into Eq. (11) yields

$$\mathcal{I}(g) = \frac{(\nabla\mathcal{L}^T \nabla\mathcal{L})^2}{\nabla\mathcal{L}^T Q \nabla\mathcal{L} + \nu^2 \sum_{i=1}^{d} \lambda_i^3}. \tag{20}$$

For stochastic sign descent, we have $\mathbf{E}[s_i] = (2\rho_i - 1)\,\text{sign}(\nabla\mathcal{L}_i)$ and thus $\nabla\mathcal{L}^T\mathbf{E}[s] = \sum_{i=1}^{d} \nabla\mathcal{L}_i \mathbf{E}[s_i] = \sum_i (2\rho_i - 1)|\nabla\mathcal{L}_i|$. Regarding the denominator, it is

$$0 \le s^T H s = |s^T Q s| = \left| \sum_{i=1}^{d} q_{ij} s_i s_j \right| \le \sum_{i=1}^{d} |q_{ij}||s_i||s_j| = \sum_{i=1}^{d} |q_{ij}|. \tag{21}$$

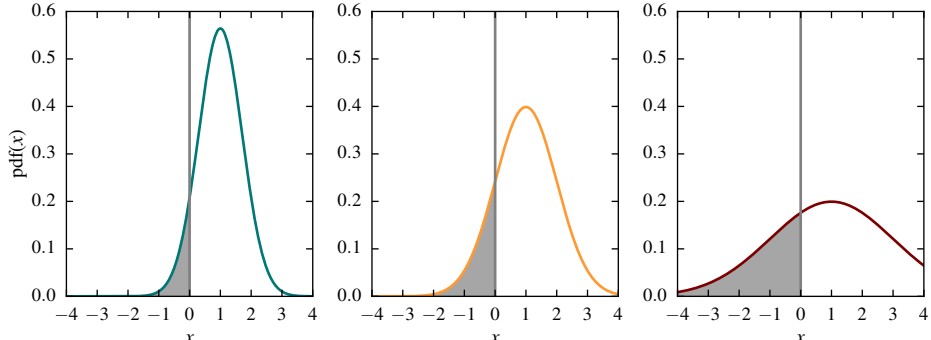

Figure 4: Probability density functions (pdf) of three Gaussian distributions, all with $\mu = 1$, but different variances $\sigma^2 = 0.5$ (left), $\sigma^2 = 1.0$ (middle), $\sigma^2 = 4.0$ (right). The shaded area under the curve corresponds to the probability that a sample from the distribution has the opposite sign than its mean. For the Gaussian distribution, this probability is uniquely determined by the fraction $\sigma/|\mu|$, as shown in Lemma 2.

Plugging everything into Eq. (11) yields

$$\mathcal{I}(s) \geq \frac{\left(\sum_{i=1}^{d}(2\rho_i - 1)|\nabla\mathcal{L}_i|\right)^2}{\sum_{i=1}^{d}|q_{ij}|}. \tag{22}$$

### B.2 SUCCESS PROBABILITIES OF THE SIGN OF A STOCHASTIC GRADIENT

We have stated in the main text that the sign of a stochastic gradient, $s(\theta) = \text{sign}(g(\theta))$, has success probabilities

$$\rho_i = \mathbf{P}[s(\theta)_i = \text{sign}(\nabla\mathcal{L}(\theta)_i)] = \frac{1}{2} + \frac{1}{2}\,\text{erf}\left(\frac{|\nabla\mathcal{L}(\theta)_i|}{\sqrt{2}\sigma(\theta)_i}\right) \tag{23}$$

under the assumption that $g \sim \mathcal{N}(\nabla\mathcal{L}, \Sigma)$. The following Lemma formally proves this statement and Figure 4 provides a pictorial illustration.

**Lemma 2.** *If $X \sim \mathcal{N}(\mu, \sigma^2)$ then*

$$\rho = \mathbf{P}[\text{sign}(X) = \text{sign}(\mu)] = \frac{1}{2}\left(1 + \text{erf}\left(\frac{|\mu|}{\sqrt{2}\sigma}\right)\right). \tag{24}$$

*Proof.* The cumulative density function (cdf) of $X \sim \mathcal{N}(\mu, \sigma^2)$ is $\mathbf{P}[X \leq x] = \Phi((x - \mu)/\sigma)$, where $\Phi(z) = 0.5(1 + \text{erf}(z/\sqrt{2}))$ is the cdf of the standard normal distribution. If $\mu < 0$, then

$$\rho = \mathbf{P}[X < 0] = \Phi\left(\frac{0 - \mu}{\sigma}\right) = \frac{1}{2}\left(1 + \text{erf}\left(\frac{-\mu}{\sqrt{2}\sigma}\right)\right). \tag{25}$$

If $\mu > 0$, then

$$\begin{aligned}
\rho = \mathbf{P}[X > 0] &= 1 - \mathbf{P}[X \leq 0] = 1 - \Phi\left(\frac{0 - \mu}{\sigma}\right) \\
&= 1 - \frac{1}{2}\left(1 + \text{erf}\left(\frac{-\mu}{\sqrt{2}\sigma}\right)\right) = \frac{1}{2}\left(1 + \text{erf}\left(\frac{\mu}{\sqrt{2}\sigma}\right)\right),
\end{aligned} \tag{26}$$

where the last step used the anti-symmetry of the error function. $\square$

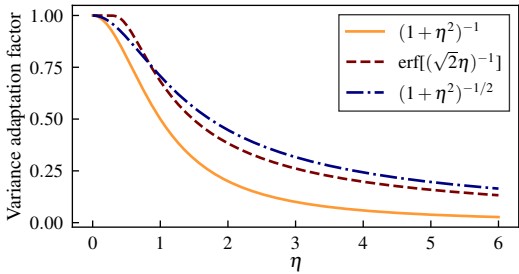

Figure 5: Variance adaptation factors as functions of the relative standard deviation $\eta$. $(1+\eta^2)^{-1}$ is the optimal variance adaptation factor for SGD (Eq. 16). The optimal factor for the sign of a stochastic gradient is $\mathrm{erf}((\sqrt{2}\eta)^{-1})$ under the Gaussian assumption (Eq. 15). It is closely approximated by $(1+\eta^2)^{-1/2}$, which is the factor implicitly employed by ADAM (Eq. 6).

### B.3 DETAILS ON VARIANCE ADAPTATION FACTORS

*Proof of Lemma 1.* Using $\mathbf{E}[\hat{p}_i] = p_i$ and $\mathbf{E}[\hat{p}_i^2] = p_i^2 + \sigma_i^2$, we get

$$
\begin{aligned}
\mathbf{E}[\|\gamma \odot \hat{p} - p\|_2^2] &= \sum_{i=1}^d \mathbf{E}[(\gamma_i \hat{p}_i - p_i)^2] = \sum_{i=1}^d \gamma_i^2 \mathbf{E}[\hat{p}_i^2] - 2\gamma_i p_i \mathbf{E}[\hat{p}_i] + p_i^2 \\
&= \sum_{i=1}^d \gamma_i^2 (p_i^2 + \sigma_i^2) - 2\gamma_i p_i^2 + p_i^2.
\end{aligned}
\tag{27}
$$

Setting the derivative w.r.t. $\gamma_i$ to zero, we find the optimal choice

$$
\gamma_i = \frac{p_i^2}{p_i^2 + \sigma_i^2}.
\tag{28}
$$

Using $\mathbf{E}[\mathrm{sign}(\hat{p}_i)] = (2\rho_i - 1)\,\mathrm{sign}(p_i)$ and $\mathrm{sign}(\cdot)^2 = 1$, we get

$$
\begin{aligned}
\mathbf{E}[\|\gamma \odot \mathrm{sign}(\hat{p}) - \mathrm{sign}(p)\|_2^2] &= \sum_{i=1}^d \gamma_i^2 \mathbf{E}[\mathrm{sign}(\hat{p}_i)^2] - 2\gamma_i \,\mathrm{sign}(p_i)\mathbf{E}[\mathrm{sign}(\hat{p}_i)] + \mathrm{sign}(p_i)^2 \\
&= \gamma_i^2 - 2\gamma_i(2\rho_i - 1) + 1
\end{aligned}
\tag{29}
$$

and easily find the optimal choice

$$
\gamma_i = 2\rho_i - 1.
\tag{30}
$$

by setting the derivative to zero. $\qquad\square$

See Figure 5 for a plot of the variance adaptation factors considered in this paper.

### B.4 CONVERGENCE OF IDEALIZED STOCHASTIC VARIANCE-ADAPTED GRADIENT

We proof the convergence results for idealized variance-adapted stochastic gradient descent. We have to clarify an aspect that we have glossed over in the main text. A stochastic optimizer generates a discrete stochastic process $\{\theta_t\}_{t\in\mathbb{N}_0}$. We denote as $\mathbf{E}_t[\cdot] = \mathbf{E}[\cdot|\theta_0, \ldots, \theta_t]$ the conditional expectation given a realization of that process up to time step $t$. Recall that $\mathbf{E}[\mathbf{E}_t[\cdot]] = \mathbf{E}[\cdot]$.

*Proof of Theorem 1.* Using the Lipschitz continuity of $\nabla f$, we can bound $f(\theta + \Delta\theta) \leq f(\theta) + \nabla f(\theta)^T \Delta\theta + \frac{L}{2}\|\Delta\theta\|^2$. Hence,

$$
\begin{aligned}
\mathbf{E}_t[f_{t+1}] &\leq f_t - \alpha \mathbf{E}_t[\nabla f_t^T(\gamma_t \odot g_t)] + \frac{L\alpha^2}{2}\mathbf{E}_t[\|\gamma_t \odot g_t\|^2] \\
&= f_t - \frac{1}{L}\sum_{i=1}^d \gamma_{t,i}\nabla f_{t,i}\mathbf{E}[g_{t,i}] + \frac{1}{2L}\sum_{i=1}^d \gamma_{t,i}^2 \mathbf{E}_t[g_{t,i}^2] \\
&= f_t - \frac{1}{L}\sum_{i=1}^d \gamma_{t,i}\nabla f_{t,i}^2 + \frac{1}{2L}\sum_{i=1}^d \gamma_{t,i}^2 (\nabla f_{t,i}^2 + \sigma_{t,i}^2).
\end{aligned}
\tag{31}
$$

Plugging in the definition

$$\gamma_{t,i} = \frac{\nabla f_{t,i}^2}{\nabla f_{t,i}^2 + \sigma_{t,i}^2} \tag{32}$$

and simplifying, we get

$$\mathbf{E}_t[f_{t+1}] \leq f_t - \frac{1}{2L} \sum_{i=1}^{d} \frac{\nabla f_{t,i}^2}{\nabla f_{t,i}^2 + \sigma_{t,i}^2} \nabla f_{t,i}^2. \tag{33}$$

Using Jensen's inequality[4]

$$\begin{aligned}
\sum_{i=1}^{d} \frac{\nabla f_{t,i}^2}{\nabla f_{t,i}^2 + \sigma_{t,i}^2} \nabla f_{t,i}^2 &= \|\nabla f_t\|^2 \sum_{i=1}^{d} \frac{\nabla f_{t,i}^2}{\|\nabla f_t\|^2} \left( \frac{\nabla f_{t,i}^2 + \sigma_{t,i}^2}{\nabla f_{t,i}^2} \right)^{-1} \\
&\geq \|\nabla f_t\|^2 \left( \sum_{i=1}^{d} \frac{\nabla f_{t,i}^2}{\|\nabla f_t\|^2} \frac{\nabla f_{t,i}^2 + \sigma_{t,i}^2}{\nabla f_{t,i}^2} \right)^{-1} \\
&= \frac{\|\nabla f_t\|^4}{\sum_{i=1}^{d}(\nabla f_{t,i}^2 + \sigma_{t,i}^2)} \geq \frac{\|\nabla f_t\|^4}{G^2}.
\end{aligned} \tag{34}$$

Due to strong convexity, we have $\|\nabla f_t\|^2 \geq 2\mu(f_t - f_*)$ and can further bound

$$\sum_{i=1}^{d} \frac{\nabla f_{t,i}^2}{\nabla f_{t,i}^2 + \sigma_{t,i}^2} \nabla f_{t,i}^2 \geq \frac{4\mu^2(f_t - f_*)^2}{G^2}. \tag{35}$$

Inserting this in (33) and subtracting $f_*$, we get

$$\mathbf{E}_t[f_{t+1}] - f_* \leq f_t - f_* - \frac{2\mu^2}{LG^2}(f_t - f_*)^2, \tag{36}$$

and, consequently, by total expectation

$$\begin{aligned}
\mathbf{E}[f_{t+1} - f_*] = \mathbf{E}\left[\mathbf{E}_t[f_{t+1}] - f_*\right] &\leq \mathbf{E}[f_t - f_*] - \frac{2\mu^2}{LG^2}\mathbf{E}[(f_t - f_*)^2] \\
&\leq \mathbf{E}[f_t - f_*] - \frac{2\mu^2}{LG^2}\mathbf{E}[f_t - f_*]^2,
\end{aligned} \tag{37}$$

which we rewrite, using the shorthand $e_t := \mathbf{E}[f_t - f_*]$, as

$$0 \leq e_{t+1} \leq e_t(1 - ce_t), \quad c = \frac{2\mu^2}{LG^2}. \tag{38}$$

To conclude the proof, we will show that this implies $e_t \in \mathcal{O}(\frac{1}{t})$. Without loss of generality, we assume $e_{t+1} > 0$ and get

$$e_{t+1}^{-1} \geq e_t^{-1}(1 - ce_t)^{-1} \geq e_t^{-1}(1 + ce_t) = e_t^{-1} + c, \tag{39}$$

where the second step is due to the simple fact that $(1-x)^{-1} \geq (1+x)$ for any $x \in [0, 1)$. Summing this inequality over $t = 0, \ldots, T-1$ yields $e_T^{-1} \geq e_0^{-1} + Tc$ and, thus,

$$Te_T \leq \left( \frac{1}{Te_0} + c \right)^{-1} \xrightarrow{T\to\infty} \frac{1}{c} < \infty, \tag{40}$$

which shows that $e_t \in \mathcal{O}(\frac{1}{t})$.  □

---

[4] Jensen's inequality says that $\sum_i c_i\phi(x_i) \geq \phi(\sum_i c_i x_i)$ for a convex function $\phi$ and convex coefficients $c_i \geq 0$, $\sum_i c_i = 1$. Here, we apply it to the convex function $\phi(x) = 1/x$, $x > 0$, and coefficients $\nabla f_{t,i}^2 / \|\nabla f_t\|^2$.

## C  MORE ON GRADIENT VARIANCE ESTIMATION

### C.1  ESTIMATES FROM MOVING AVERAGES

Iterating the recursive formula for $\tilde{m}_t$ backwards, we get

$$m_t = \frac{\tilde{m}_t}{1 - \beta_1^t} = \frac{1}{1 - \beta_1^t}\left(\beta_1 m_{t-1} + (1 - \beta_1)g_t\right) = \ldots = \frac{1 - \beta_1}{1 - \beta_1^t}\sum_{s=0}^{t-1}\beta_1^s g_{t-s}. \tag{41}$$

Hence, $m_t$ is a weighted average of past observed gradients with coefficients $c(\beta_1, t, s) := \beta_1^s(1 - \beta_1)/(1 - \beta_1^t)$, which sum to one, since $\sum_{s=0}^{t-1}\beta_1^s = (1 - \beta_1^t)/(1 - \beta_1)$ by the geometric sum formula. The analogous statement holds for $v_t$. The basic rationale that facilitates a variance estimate from past gradient observation is to assume that the true gradient does not change drastically over the effective time horizon of the exponential moving average. For mathematical simplicity, we can translate this assumption to mean that, at the $t$-th step, we treat all $\{g_{t-s,i} \mid s = 0, \ldots, t-1\}$ as iid with mean $\nabla\mathcal{L}_{t,i}$ and variance $\sigma_{t,i}^2$. This will of course be utterly wrong for gradient observations that are far in the past, but since $c(\mu, t, s)$ is very small for large $t - s$, these won't contribute significantly to the moving average. The moving average constant defines the effective time horizon, for which we implicitly make this assumption.

Under this peculiar assumption, $m_t$ and $v_t$ are unbiased estimates of the first and second moment of $g_t$, respectively:

$$\mathbf{E}[m_{t,i}] = \sum_{s=0}^{t-1}c(\mu, t, s)\mathbf{E}[g_{t-s,i}] = \nabla\mathcal{L}_{t,i}\sum_{s=0}^{t-1}c(\mu, t, s) = \nabla\mathcal{L}_{t,i}, \tag{42}$$

$$\mathbf{E}[v_{t,i}] = \sum_{s=0}^{t-1}c(\mu, t, s)\mathbf{E}[g_{t-s,i}^2] = (\nabla\mathcal{L}_{t,i}^2 + \sigma_{t,i}^2)\sum_{s=0}^{t-1}c(\mu, t, s) = \nabla\mathcal{L}_{t,i}^2 + \sigma_{t,i}^2, \tag{43}$$

motivating $v_t - m_t^2$ as a gradient variance estimate. However, $v_t - m_t^2$ is not an unbiased variance estimate due to the fact $m_t^2$ is not an unbiased estimate of $\nabla\mathcal{L}_t^2$. The error arising from this bias should generally be dominated by other error sources and will thus be ignored.

### C.2  MINI-BATCH ESTIMATES

An alternative gradient variance estimate can be obtained locally, within a single mini-batch. The individual gradients $\nabla\ell(\theta, x_k)$ in a mini-batch are iid random variables and, as noted in the introduction, $\mathbf{var}[g(\theta)] = |\mathcal{B}|^{-1}\mathbf{var}[\nabla\ell(\theta, x_k)]$. We can thus estimate $g(\theta)$'s variances by computing the sample variance of the $\{\nabla\ell(\theta, x_k)\}_{k\in\mathcal{B}}$, then scaling by $|\mathcal{B}|^{-1}$,

$$\hat{s}(\theta) = \frac{1}{|\mathcal{B}|}\left(\frac{1}{|\mathcal{B}| - 1}\sum_{k\in\mathcal{B}}\nabla\ell(\theta, x_k)^2 - g(\theta)^2\right). \tag{44}$$

Several recent papers (Mahsereci & Hennig, 2015; Balles et al., 2017b; Mahsereci et al., 2017) have used this variance estimate for other aspects of stochastic optimizers. In contrast to $v_t - m_t^2$, this is an unbiased estimate of the *local* gradient variance. The (non-trivial) implementation of this estimator for neural networks is described in Balles et al. (2017a).

### C.3  RELATIVE VARIANCE OF A MOMENTUM TERM (DERIVATION OF EQ. 19)

When estimating the variance with moving averages, we assume that $\mathbf{E}[g_t] = m_t$ and $\mathbf{var}[g_t] = v_t - m_t^2$. Plugging this into Eq. (18) we can approximate the mean and variance of the momentum term by

$$\mathbf{E}[r_t]^2 \approx \left(\sum_{s=0}^{t}\mu^s m_{t-s}\right)^2, \quad \mathbf{var}[r_t] \approx \sum_{s=0}^{t}\mu^{2s}(v_{t-s} - m_{t-s}^2). \tag{45}$$

Computing these two expressions would require two more moving averages in addition to $m_t$ and $v_t$. However, $m_t$ and $v_t$ will change slowly over time and, by using $v_t - m_t^2$ as the variance estimate for

$g_t$, we anyways make the assumption that all gradients in the effective time horizon of the moving average have the same mean and variance. We thus further approximate by replacing $m_{t-s}$ with $m_t$ and get

$$\mathbf{E}[r_t]^2 \approx m_t^2 \left( \sum_{s=0}^{t} \mu^s \right)^2 = m_t^2 \left( \frac{1 - \mu^t}{1 - \mu} \right)^2, \tag{46}$$

$$\mathbf{var}[r_t] \approx (v_t - m_t^2) \sum_{s=0}^{t} \mu^{2s} = (v_t - m_t^2) \frac{1 - \mu^{2t}}{1 - \mu^2}. \tag{47}$$

The two scalar factors lead to the correction term $\kappa(\mu, t)$ in Eq. (19).

When estimating the gradient variance from the mini-batch (Eq. 44), we can obtain an unbiased estimate of $\mathbf{var}[r_t]$ in Eq. (18) via

$$\bar{s}_t = \mu^2 \bar{s}_{t-1} + \hat{s}_t, \tag{48}$$

where $\hat{s}_t$ is given by Eq. (44).

## D  VARIATIONS OF VARIANCE-ADAPTED METHODS

Based on the considerations in Section 3, we examined three more variance-adapted methods. The first is a variation of M-SVAG which estimates stochastic gradient variances locally within the mini-batch, as explained in §C.2. Pseudo-code can be found in Alg. 5. Furthermore, we tested a variant of ADAM that applies the correction factor from Eq. (19) to the estimate of the relative variance of the momentum term. We refer to this method as ADAM*. Two variants of ADAM* with the two variance estimates can be found in Algorithms 4 and 5.

---

**Algorithm 3** M-SVAG-mb (with mini-batch variance estimates)

---

**Require:** initial value $\theta_0$, step size $\alpha$, momentum parameter $\mu \in [0, 1]$, number of steps $T$

1: Initialize $m = 0$, $\bar{s} = 0$
2: **for** $t = 1, \dots, T$ **do**
3:     Compute stochastic gradient $g(\theta)$ and variance estimate $\hat{s}(\theta)$     ▷ Eq. (44)
4:     Update aggregators $m \leftarrow \mu m + g(\theta)$,     $\bar{s} \leftarrow \mu^2 \bar{s} + \hat{s}(\theta)$
5:     Compute relative variance estimate $\eta^2 = \bar{s}/m^2$
6:     Compute variance adaptation factors $\gamma = (1 + \eta^2)^{-1}$
7:     Update $\theta \leftarrow \theta - \alpha(\gamma \odot m)$
8: **end for**

---

**Algorithm 4** ADAM* (with exp. moving average variance estimates)

---

**Require:** initial value $\theta_0$, step size $\alpha$, momentum/averaging constant $\mu \in [0, 1]$, number of steps $T$

1: Initialize $m = 0$, $v = 0$
2: **for** $t = 1, \dots, T$ **do**
3:     Compute stochastic gradient $g = g(\theta)$
4:     Update moving averages $m \leftarrow \mu m + (1 - \mu)g$,     $v \leftarrow \mu v + (1 - \mu)g^2$
5:     Bias-correct $m = (1 - \mu^t)^{-1}\tilde{m}$,     $v = (1 - \mu^t)^{-1}\tilde{v}$
6:     Compute relative variance estimate $\eta^2 = \kappa(\mu, t)\frac{v - m^2}{m^2}$     ▷ Eq. (19)
7:     Compute variance adaptation factors $\gamma = (1 + \eta^2)^{-1/2}$
8:     Update $\theta \leftarrow \theta - \alpha(\gamma \odot \text{sign}(m))$
9: **end for**

This is ADAM ($\beta_1 = \beta_2 = \mu$, $\varepsilon = 0$), expect for the correction factor $\kappa(\mu, t)$ for the relative variance.

---

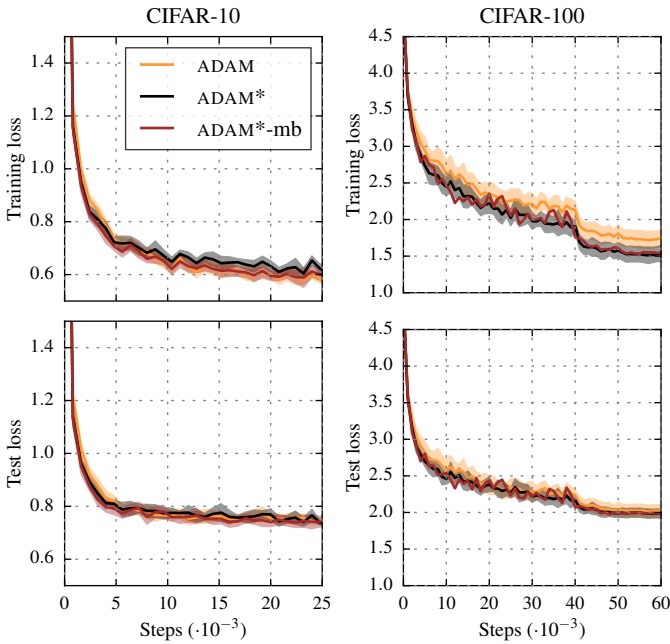

Figure 6: Comparison of the original ADAM algorithm to the variants in Algs. 4 and 5. Set-up of the plots as in Fig. 3. All three algorithms exhibit very similar performance on both problems.

---

**Algorithm 5** ADAM*-mb (with mini-batch variance estimates)

---

**Require:** initial value $\theta_0$, step size $\alpha$, momentum/averaging constant $\mu \in [0, 1]$, number of steps $T$

1: Initialize $m = 0$, $\bar{s} = 0$
2: **for** $t = 1, \dots, T$ **do**
3:     Compute stochastic gradient $g(\theta)$ and variance estimate $\hat{s}(\theta)$            $\triangleright$ Eq. (44)
4:     Update aggregators $m \leftarrow \mu m + g(\theta)$,     $\bar{s} \leftarrow \mu^2 \bar{s} + \hat{s}(\theta)$
5:     Compute relative variance estimate $\eta^2 = \bar{s}/m^2$
6:     Compute variance adaptation factors $\gamma = (1 + \eta^2)^{-1/2}$
7:     Update $\theta \leftarrow \theta - \alpha(\gamma \odot \text{sign}(m))$
8: **end for**

---

### D.1 EXPERIMENTAL RESULTS

We evaluated the variants on the two CIFAR test problems. Figure 6 shows a comparison of the two ADAM* variants with the original ADAM. Figure 7 compares the mini-batch variant of M-SVAG to the one with exponential moving averages.

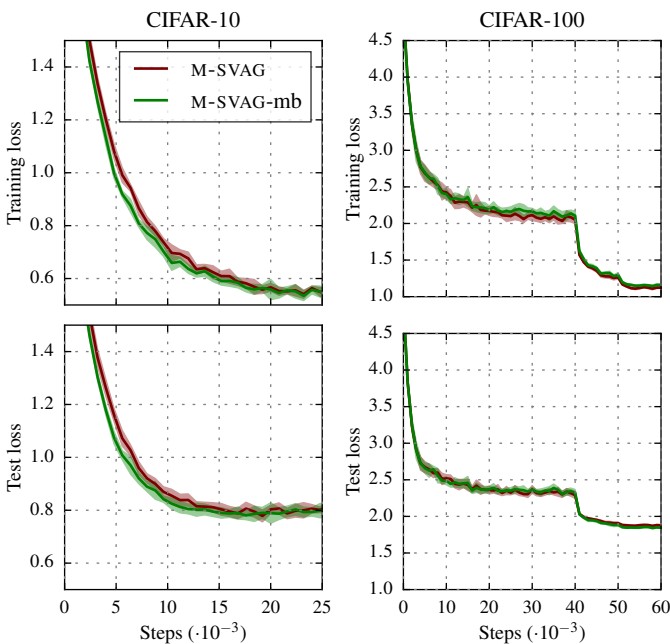

Figure 7: Comparison of the two variants of the M-SVAG algorithm. Set-up of the plots as in Fig. 3. Both variants exhibit very similar performance on both problems.

