# OpenReview forum: "Dissecting Adam: The Sign, Magnitude and Variance of Stochastic Gradients"
_ICLR.cc/2018/Conference — Reject_

### Official Review · AnonReviewer2 · 2017-11-25
**Dissecting Adam: The Sign, Magnitude and Variance of Stochastic Gradients**

**Rating:** 4
**Confidence:** 4

**Review:**

Summary:
The paper is trying to improve Adam based on variance adaption with momentum. Two algorithms are proposed, M-SSD (Stochastic Sign Descent with Momentum) and M-SVAG (Stochastic Variance-Adapted Gradient with Momentum) to solve finite sum minimization problem. The convergence analysis is provided for SVAG for strongly convex case. Numerical experiments are provided for some standard neural network structures with three common datasets MNIST, CIFAR10 and CIFAR100 compared the performance of M-SSD and M-SVAG to two existing algorithms: SGD momentum and Adam.

Comments:
Page 4, line 5: You should define \nu clearly.

Theorem 1: In the strongly convex case, assumption E ||g_t ||^2 \leq G^2 (if G is a constant) is too strong. In this case, G could be equal to infinity. If G is not infinity, you already assume that your algorithm converges, that is the reason why this assumption is not so good for strongly convex. If G is infinity (this is really possible for strongly convex), your proof would get a trouble as eq. (40) is not valid anymore.

Also, to compute \gamma_{t,i}, it requires to compute \nabla f_{t,i}, which is full gradient. By doing this, the computational cost should add the dependence of M, which is very large as you mentioned in the introduction. According to your rate O(1/t), the complexity is worse than that of gradient descent and SGD as well.

As I understand, there is no theoretical results for M-SSG and M-SVAG, but only the result for SVAG with exact \eta_i^2 in the strongly convex case. Also, theoretical results are not strong enough. Hence, the experiments need to make more convincingly, at least for some different complicated architecture of deep neural network. As I see, in some dataset, Adam performs better than M-SSD, some another dataset, Adam performs better than M-SVAG. Same situation for M-SGD. My question is that: When should we use M-SSD or M-SVAG? For a given dataset, why should we not use Adam or M-SGD (or other existing algorithms such as Adagrad, RMSprop), but your algorithms?

You should do more experiments to various dataset and architectures to be more convincing since theoretical results are not strong enough. Would you think to try to use VGG or ResNet to ImageNet?

I like the idea of the paper but I would love if the author(s) could improve more theoretical results to convince people. Otherwise, the results in this paper could not be considered as good enough. At this moment, I think the paper is still not ready for the publication.

Minor comments:
Page 2, in eq. (6): You should mention that “1” is a vector.
Page 4, line 4: Q in R^{d} => Q in R^{d x d}
Page 6, Theorem 1: You should define the finite sum optimization problem with f since you have not used it before.
Page 6, Theorem 1: You should use another notation for “\mu”-strongly convex parameter since you have another “\mu”-momentum parameter in section 3.4
Page 4, Page 7: Be careful with the case when c = 0 (page 4) and mu = 1 (page 7-8) with dividing by 0.

---

> ### Author Response · Authors · 2017-12-07
> **Response to the Comments of Reviewer 2**
>
> Dear Reviewer 2,
>
> thanks for your constructive review. We want to address multiple of the points you have raised.
>
> 1) "The paper is trying to improve Adam based on variance adaption with momentum. Two algorithms are proposed, M-SSD (Stochastic Sign Descent with Momentum) and M-SVAG (Stochastic Variance-Adapted Gradient with Momentum) to solve finite sum minimization problem."
>
> We do not want to improve Adam, but provide insight into its inner workings. We argue that Adam is a combination of two aspects (sign-based and variance adaptation), which can be separated. M-SSD and M-SVAG are the results of this separation. We do not propose M-SSD as a method to be used in practice, but merely as a baseline for comparison (we comment on this in more detail below).
>
> 2) "Theorem 1: In the strongly convex case, assumption E ||g_t ||^2 \leq G^2 (if G is a constant) is too strong. In this case, G could be equal to infinity. If G is not infinity, you already assume that your algorithm converges, that is the reason why this assumption is not so good for strongly convex. If G is infinity (this is really possible for strongly convex), your proof would get a trouble as eq. (40) is not valid anymore."
>
> As we already have pointed out in our response to Reviewer 1, this is a standard assumption in convergence proofs of stochastic optimization methods. To name just a few examples: [1] (Theorem 1), [2] (Theorem 4.1), [3] (e.g. Theorem 4).  It does *not* assume that the algorithm convergences, only that it does not diverge ($|| \nabla f_t ||^2 < \infty$) and that the noise is bounded ($\sum_i \sigma(\theta)_i^2 < \infty$ for all $\theta$). We can add a clarifying remark to the paper.
>
> 3) "Also, to compute \gamma_{t,i}, it requires to compute \nabla f_{t,i}, which is full gradient. By doing this, the computational cost should add the dependence of M, which is very large as you mentioned in the introduction. According to your rate O(1/t), the complexity is worse than that of gradient descent and SGD as well."
>
> We are providing a theoretical result for *idealized* SVAG, where we assume access to the exact relative variance. As we write in the paper, this is meant as a *motivation* for this form of variance adaptation. We do not provide a theoretical result for SVAG (or M-SVAG) with *estimated* variances, but evaluate these methods empirically.
>
> 4) "My question is that: When should we use M-SSD or M-SVAG? For a given dataset, why should we not use Adam or M-SGD (or other existing algorithms such as Adagrad, RMSprop), but your algorithms?"
>
> While our analysis in Section 2 provides some insight into when to use sign-based methods over SGD, we don't have a conclusive answer to your question. But couldn't we ask the same question for any other optimization method used in Deep Learning? Why should we use Adam instead of SGD+momentum? The answer is that it has been shown empirically to work better on some (but by no means all) problems. In this work, we show empirically that M-SVAG consistently improves over standard SGD with momentum. It would thus be a logical choice on problems where SGD+momentum outperforms Adam. We specifically avoided to "sell" M-SVAG as "the new optimizer that everybody should use now". It is an addition to the toolbox; it uses variance-based element-wise step sizes, but is not based on the sign of the gradient. Regarding M-SSD, we want to point out that we do not see this as a method to be used in practice. We included it in the comparison for completeness, since it adopts the sign aspect of Adam but removes the variance adaptation (see Table 1). We will make this more clear in a revised version of the paper.
>
> 5) "You should do more experiments to various dataset and architectures to be more convincing since theoretical results are not strong enough. Would you think to try to use VGG or ResNet to ImageNet?"
>
> We agree that more experimental results are always better. However, this is very computationally demanding. With the individual learning rate tuning for each method and 10 replication runs, adding a new data set / architecture amounts to roughly 100 training runs. (There are enough papers out there that make claims about an optimization method based on a single run with a single learning rate, but we do not want to do that.)
>
> We hope we were able to alleviate some of your concerns. We kindly ask you to reconsider your evaluation of the paper in light of this response.
>
> [1] Shamir and Zhang. Stochastic Gradient Descent for Non-smooth Optimization: Convergence Results and Optimal Averaging Schemes. 2012.
> [2] Kingma and Ba. Adam: A Method for Stochastic Optimization. 2015.
> [3] Karimi et al. Linear Convergence of Gradient and Proximal-Gradient Methods Under the Polyak- Lojasiewicz Condition, 2016.

---

> > ### Comment · AnonReviewer2 · 2017-12-08
> > **Response to the author(s)**
> >
> > Dear author(s),
> >
> > Thank you for your response!
> >
> > 2) "Theorem 1: In the strongly convex case, assumption E ||g_t ||^2 \leq G^2 (if G is a constant) is too strong. In this case, G could be equal to infinity. If G is not infinity, you already assume that your algorithm converges, that is the reason why this assumption is not so good for strongly convex. If G is infinity (this is really possible for strongly convex), your proof would get a trouble as eq. (40) is not valid anymore."
> >
> > As we already have pointed out in our response to Reviewer 1, this is a standard assumption in convergence proofs of stochastic optimization methods. To name just a few examples: [1] (Theorem 1), [2] (Theorem 4.1), [3] (e.g. Theorem 4).  It does *not* assume that the algorithm convergences, only that it does not diverge ($|| \nabla f_t ||^2 < \infty$) and that the noise is bounded ($\sum_i \sigma(\theta)_i^2 < \infty$ for all $\theta$). We can add a clarifying remark to the paper.
> >
> > RE: It is true that this is a standard assumption in convergence proofs of many previous papers. However, this is also well-known that for strongly convex case, this assumption is not valid. The reason is as follows. Your f(\theta) = E[f_i(\theta)] is \mu-strongly convex and L-smooth. By \mu-strongly convex property of f, we have: for all \theta \in R^d
> >
> > 2*\mu*[f(\theta) - f(\theta*)] \leq || \nabla f(\theta) ||^2 = || E[\nabla f_i(\theta)] ||^2 \leq E || \nabla f_i(\theta) ||^2 \leq G (according to your assumption)
> >
> > From here, we have f(\theta) \leq G/(2*\mu) + f(\theta*) = FIXED CONSTANT for all \theta \in R^d since \theta* is a unique solution of strongly convex f and \mu and G are also fixed (by your assumption). This means that you implicitly assume that your algorithm converges in some FIXED neighborhood. Although this neighborhood is large, there is no guarantee that your algorithm never goes out of this region. Please notice that assuming || \nabla f_t ||^2 < \infty and || \nabla f_t ||^2 \leq G < \infty are TOTALLY different since you are fixing G. If you allow G becomes arbitrary large (G -> \infty), then your algorithm would be fine. But like I said before, if G -> infinity, your proof would get a trouble since eq. (40) is not valid anymore.
> >
> > I know that many previous papers were assuming this assumption (when G is fixed) with strong convexity. But this is only true for an empty class function satisfying both conditions.
> >
> > In my opinion, your theoretical result is not rigorous enough.

---

> > > ### Author Response · Authors · 2017-12-08
> > > **Re: Assumption in Theorem 1**
> > >
> > > Dear Reviewer 2,
> > >
> > > You are misunderstanding the assumption. We do _not_ assume that E[ || \nabla g(\theta) ||^2 ] \leq G^2 for all \theta in R^d. That would of course be utterly wrong. We assume that E [ || \nabla g(\theta_t) ||^2 ] \leq G^2 for all t, i.e., the expected squared norm of stochastic gradients __at the iterates of the algorithm__ are bounded uniformly! This assumption is made to bound the variance of stochastic gradients (it sometimes referred to as the finite variance condition). The (non-stochastic) gradient norms || \nabla f(\theta_t) ||^2 are bounded since the algorithm only explores a bounded region of the search space, as a direct consequence of its non-divergence. (This follows immediately from Eq. (33), which says that we expect a descent in function value at each step; if you insist, I can write it up and add it to the paper.)
> > >
> > > Admittedly, assuming E [ || \nabla g(\theta_t) ||^2 ] \leq G^2 entangles these two things. However, it is a standard and absolutely valid assumption, which keeps the proof concise and readable. Here are some more papers that use it, either in exactly this form, or in slight variations:
> > > [4] Assumtption (c) on the first page
> > > [5] Theorem 2.3
> > > [6] Assumption 3 (a) in Section 2.1
> > > [7] Assumption 4.3
> > > If you insist that this assumption is invalid, you are questioning a good portion of research on stochastic optimization methods.
> > >
> > > Irrespective of our disagreement, thanks for checking the proof and engaging in this conversation!
> > >
> > >
> > > [4] Simon Lacoste-Julien, Mark Schmidt, Francis Bach. A simpler approach to obtaining an O(1/t) convergence rate for the projected stochastic subgradient method. 2012.
> > > [5] Michael Friedlander, Mark Schmidt. Hybrid Deterministic-Stochasic Methods for Data Fitting. 2011.
> > > [6] Elad Hazan, Satyen Kale. Beyond the Regret Minimization Barrier: Optimal Algorithms for Stochastic Strongly-Convex Optimization. 2014.
> > > [7] Leon Bottou, Frank Curtis, Jorge Nocedal. Optimization Methods for Large-Scale Machine Learning. 2016.

---

> > > > ### Comment · AnonReviewer2 · 2017-12-08
> > > > **Assumption**
> > > >
> > > > Dear author(s),
> > > >
> > > > I just provided the expressions to show that it is not true for all \theta in R^d. I clearly stated for your case.
> > > >
> > > > "From here, we have f(\theta) \leq G/(2*\mu) + f(\theta*) = FIXED CONSTANT for all \theta \in R^d since \theta* is a unique solution of strongly convex f and \mu and G are also fixed (by your assumption). This means that you implicitly assume that YOUR ALGORITHM converges in some FIXED neighborhood"
> > > >
> > > > I undertand that you are assuming for all t for your algorithm. But this means that you are assuming all f(\theta_1), ... , f(\theta_t) are ALWAYS smaller than some "FIXED particular constant" since you fix G and mu and of course f(\theta*) is also fixed. There is no guarantee that your updates is always in this fixed region since you are considering \theta_1, ... , \theta_t \in R^d. What if at some time \tau, there exists a f(\theta_\tau) which is greater than that fixed constant? This is always possible since you are not limited your updates. You cannot just simply assume that those updates are always in that fixed region. This implicity implies that you are assuming your algorithm converges in some fixed region (which is not true since they may go out).
> > > >
> > > > For your references, they should add something else for the assumption. For example, in [1], they projected their updates into some convex set (this should be bounded convex set). For [2], [3], you can see they provide more supports to that assumption, either considering problems in some convex bounded set or adding something else. For [4], E || \nabla f_i(\theta_t) ||^2 \leq M + N* ||\nabla f(\theta_t) ||^2, this is somehow still reasonable since you are not limited the RHS, which means they are still allowing ||\nabla f(\theta_t)||^2 become arbitrary large.
> > > >
> > > > Anyway, I know some specific papers assuming that assumption with fixed G in the strongly convex case. But like I said before, this is only true for an empty class function. To correct your proof, you should allow G become arbitrary large, which means allowing G -> \infty. But like I said, in this case, in your proof, c -> 0 and eq (40) is not true anymore since 1/c -> \infty.
> > > >
> > > > In other words, you can assume E || g_t ||^2 < \infty but not E ||g_t||^2 \leq G < \infty for some fixed G.

---

### Official Review · AnonReviewer1 · 2017-11-26

**Rating:** 4
**Confidence:** 4

**Review:**

The paper presents some analysis of the scale-invariance and the particular shape of the learning rate used in Adam. The paper argues that Adam's update is a combination of a sign-update and a variance-based learning rate. Some analysis is provided on these two aspects.

After spending a sizeable amount of time with this paper, I am not sure what are its novel contributions and why it should be published in a scientific conference. The paper contains so many approximations, simplifications, and assumptions that make any presented result extremely weak.

More in details, the analysis of the sign is done in the case of quadratic functions of Gaussian variables. The result is mildly interesting, but I fail to see how this would give us a hint of what is happening during the minimization of the non-convex objectives for training deep networks.
Moreover, the analysis of sign based updates has been already carried over using the Polyak-Łojasiewicz assumption in Karimi et al. ECML-PKDD 2016, that is strictly more general than any quadratic approximation.

The similarity between the ``optimal'' variance-based learning rate and the one of Adam hinges again on the fact that the noise is Gaussian. As the authors admit, Schaul et al. (2013) already derived similar updates. Also, Theorem 1 recover the usual rate of convergence for strongly convex function: How is this theorem supposed to support the fact that variance-adapted learning rates are a better idea than the usual updates?
Moreover, the proof of Theorem 1 hinges on the fact that E[||g_t||^2]\leq G^2. Clearly, this is not possible in general for a strongly convex function. The proof might still go through, but it needs to be fixed using the fact that the updates always decrease the function.

Overall, if we are considering only the convex case, Adam is clearly sub-optimal from all the points of view and better algorithms with stronger guarantees can be used. Indeed, the fact that non-convexity is never discussed is particularly alarming. It is also indicative that none of the work for minimization of finite sums are cited or discussed, e.g. the variance reduced methods immediately come to mind.

Regarding the experiments, the parameters are chosen to have the best test accuracy, mixing the machine learning problem with the optimization one: it is well-known and easy to prove that a worst optimizer can give rise to better test errors. Hence, the empirical results cannot be used to support any of the proposed interpretations nor the new optimization algorithms.

To summarize, I do not think the contributions of this paper are enough to be published in ICLR.

---

> ### Author Response · Authors · 2017-12-07
> **Response to the Comments of Reviewer 1**
>
> Dear Reviewer 1,
> thank you for your constructive review. We want to address some of the concerns you have raised.
>
> 1) "the analysis of the sign is done in the case of quadratic functions of Gaussian variables. [...] I fail to see how this would give us a hint of what is happening during the minimization of the non-convex objectives for training deep networks."
>
> We disagree with this point. Studying the behavior of optimization algorithms on simple problems is insightful, helps direct future research and, thus, is an important step to gain a deeper understanding of the method on more complex problems. The chosen problem class is simple but non-trivial, allowing to study the interplay of stochasticty and curvature. Our analysis adds insight about the effects of noise and curvature on the sign-based method and SGD.
>
> 2) "the analysis of sign based updates has been already carried over using the PL assumption in Karimi et al. [...], that is strictly more general than any quadratic approximation."
>
> Karimi et al. [3] provide a convergence proof for a sign-based method under the PL assumption and derive a general worst-case rate. We are asking a different question: In what specific situations (curvature properties and noise) can the sign direction outperform SGD? Also, [3] only consider sign-based methods in the *noise-free* case, whereas we specifically analyze the interplay of noise and malicious curvature. We will include a pointer to [3] in a revised version of the paper, but this does not affect the significance of this work.
>
> 3) "The similarity between the ``optimal'' variance-based learning rate and the one of Adam hinges on the fact that the noise is Gaussian."
>
> Since a mini-batch stochastic gradient is the mean of individual per-training-example gradients (iid random variables), the Gaussian assumption is (asymptotically) supported by the CLT. We have done some qualitative experiments on this; stochastic gradients are not perfectly Gaussian, but it is a reasonable approximation at commonly-used mini-batch sizes. We are happy to include these experiments in the supplements.
>
> 4) "Theorem 1 recovers the usual rate of convergence for strongly convex function: How is this theorem supposed to support the fact that variance-adapted learning rates are a better idea than the usual updates?"
>
> It does not improve the rate, but it achieves it without an "external" decrease of the learning rate (e.g. a global 1/t schedule). Also, SVAG locally leads to a larger expected decrease than SGD (leading to better constants in the O(1/t) rate). This fact is currently hidden away in the proof, but becomes clear when we see that our choice for \gamma_t minimizes the rhs of the first line of Eq (31).
>
> 5) "Moreover, the proof of Theorem 1 hinges on the fact that E[||g_t||^2]\leq G^2. Clearly, this is not possible in general for a strongly convex function."
>
> This is a standard assumption in convergence proofs of stochastic optimization methods. To name just a few examples: [1] (Theorem 1), [2] (Theorem 4.1), [3] (e.g. Theorem 4). It assumes that the algorithm does not diverge and that the noise is bounded. We can add a clarifying remark to the paper. As you point out, the non-divergence of the algorithm could be established in a first step.
>
> 6) "It is also indicative that none of the work for minimization of finite sums are cited or discussed, e.g. the variance reduced methods immediately come to mind."
>
> Variance-reduced methods are orthogonal to our work. They aim to construct gradient estimates with a lower variance, whereas we assume a gradient estimate as given and try to "manage" the variance by adapting per-element step sizes. These two approaches could be combined in future work. We will point out this connection in a revised version of the related work section.
>
> 7) "Regarding the experiments, the parameters are chosen to have the best test accuracy, mixing the machine learning problem with the optimization one"
>
> We agree that this is a problem. Ironically, on a different paper, where we treated NN training purely as an optimization problem, we had reviewers complain, asking for a test-set-based comparison. The community hasn't agreed on standard procedures for this. Do you have any recommendations how to please both sides? We can include a comparison based purely on train loss in the supplements.
>
>
> We think that we have addressed your main concerns, especially about the assumption in Theorem 1 and the relationship to other works (variance-reduced methods, Karimi et al. [3]). We would thus like to ask you to reconsider your evaluation in light of this response.
>
> [1] Shamir and Zhang. Stochastic Gradient Descent for Non-smooth Optimization: Convergence Results and Optimal Averaging Schemes. 2012.
> [2] Kingma and Ba. Adam: A Method for Stochastic Optimization. 2015.
> [3] Karimi et al. Linear Convergence of Gradient and Proximal-Gradient Methods Under the Polyak- Lojasiewicz Condition, 2016.

---

### Official Review · AnonReviewer3 · 2017-11-28
**The paper splits ADAM algorithm into two components: stochastic direction in sign of gradient and adaptive stepwise with relative variance. Two algorithms are proposed to test each of them. For MNIST, CIFAR10, and CIFAR100 datasets, Stochastic Sign Descent shows some better performance than others in the former two but worse in the last one.**

**Rating:** 6
**Confidence:** 3

**Review:**

Stochastic Sign Descent (SSD) and Stochastic Variance Adapted Gradient (SVAG) are inspired by ADAM and studied in this paper, together with momentum terms.

Analysis showed that SSD should work better than usual SGD when the Hessian of training loss is highly diagonal dominant.  It is intrigued to observe that for MNIST and CIFAR10, SSD with momentum champions with better efficiency than ADAM, SGD and SVAG, while on the other hand, in CIFAR100, momentum-SVAG and SGD beat SSD and ADAM. Does it suggest the Hessians associated with MNIST and CIFAR10 training loss more diagonally dominant?

There are other adaptive step-sizes such as Barzilai-Borwein (BB) Step Sizes introduced to machine learning by Tan et al. NIPS 2016. Is there any connections between variance adaptation here and BB step size?

---

> ### Author Response · Authors · 2017-12-08
> **Response to the Comments of Reviewer 3**
>
> Dear Reviewer 3,
>
> thank you for your positive review!
>
> "Does it suggest the Hessians associated with MNIST and CIFAR10 training loss more diagonally dominant?"
>
> Our analysis in Section 2 suggests this (and its interplay with stochasticity) as a possible explanation. It would be an interesting addition to the paper to investigate the diagonal dominance empirically on these two problems. However, since these are very high-dimensional problems, this would require serious additional effort (computationally and implementation). We think that this is beyond the scope of this conference paper.
>
> "Is there any connections between variance adaptation here and BB step size?"
>
> The BB is a scalar step size, whereas we suggest manipulating the per-element update magnitudes, thereby altering the update direction. Also, the BB step size arises from "geometric" considerations (the secant equation) in the noise-free case, whereas we aim to control the effect of stochasticity. So, to answer your question, I don't think that there is a relevant connection. My understanding is that the BB step size is agnostic of the search direction, so it could even be combined with our variance-adapted update direction.

---

### Author Response · Authors · 2017-11-03
**Typo**

Dear Reviewers, we want to point out that there is an unfortunate typo in the line after Eq. (9). The covariance matrix of $g(\theta)$ should obviously be $Q (\nu^2 I) Q^T = \nu^2 QQ$ instead of $\nu^2 I$. It's purely a typo; the subsequent considerations use the correct covariance matrix (see e.g., Section B.1).

---

### Decision · Program_Chairs · 2018-01-29
**ICLR 2018 Conference Acceptance Decision**

**Decision:**

Reject

**Comment:**

This paper presents a theoretical justification for the Adam optimizer in terms of decoupling the signs and magnitudes of the gradients. The overall analysis seems reasonable, though there's been much back-and-forth with the reviewers about particular claims and assumptions. Overall, the contributions don't feel quite substantial enough for an ICLR publication. The interpretation in terms of signs is interesting, but it's very similar to the motivation for RMSprop, of which Adam is an extension. The performance result on diagonally dominant noisy quadratics is interesting, but it feels unsurprising that a diagonal curvature approximation would work well in this setting. I don't recommend acceptance at this point, though these ideas could potentially be developed further into a strong submission.